# FactoredRL: Leveraging Factored Graphs for Deep Reinforcement Learning

## Abstract

We propose a simple class of deep reinforcement learning (RL) methods, called FactoredRL, that can leverage factored environment structures to improve the sample efficiency of existing model-based and model-free RL algorithms. In tabular and linear approximation settings, the factored Markov decision process literature has shown exponential improvements in sample efficiency by leveraging factored environment structures. We extend this to deep RL algorithms that use neural networks. For model-based algorithms, we use the factored structure to inform the state transition network architecture and for model-free algorithms we use the factored structure to inform the Q network or the policy network architecture. We demonstrate that doing this significantly improves sample efficiency in both discrete and continuous state-action space settings.

## 1 Introduction

In many domains, the structure of the Markov Decision Process (MDP) is known at the time of problem formulation. For example, in inventory management, we know the structure of the state transition: how inventory flows from a vendor, to a warehouse, to a customer (Giannoccaro & Pontrandolfo, 2002; Oroojlooyjadid et al., 2017). In portfolio management, we know that a certain asset changes only when the agent buys or sells a corresponding item (Jiang et al., 2017). Similar structural information is available in vehicle routing, robotics, computing, and many others. Our work stems from the observation that we can exploit the known structure of a given MDP to learn a good policy. We build on the Factored MDP literature (Boutilier et al., 1995; Osband & Van Roy, 2014; Kearns & Singh, 2002; Cui & Khardon, 2016), and propose a *factored graph* to represent known relationships between states, actions and rewards in a given problem. We use the factored graphs to inform the structure of the neural networks used in deep reinforcement learning (RL) algorithms to improve their sample efficiency. We give literature references and example factor graphs for real world applications in Appendix A.

Consider a motivational example, where the goal of the agent is to balance multiple independent cartpoles simultaneously, with each cartpole defined as per OpenAI gym (G. Brockman & Zaremba, 2016). The agent can take a 'left' or 'right' action on each cartpole, and the state includes the position and velocity of each cart and each pole. We refer to this as the Multi-CartPole problem.

Both model-based and model-free algorithms treat the state-action space as a single entity, which makes exploration combinatorially complex. As a consequence, the sample efficiency of RL algorithms degrades exponentially with the number of cartpoles, despite the problem remaining conceptually simple for a human. By allowing the agent access to the problem's factored structure (i.e. each action affects only one cartpole), we bypass the need to learn about each action's relationship with the entire state, and instead only need to learn about each action's relationship with its single, related cartpole.

We show how to integrate knowledge of the factored graph into both model-based and model-free deep RL algorithms, and thereby improve sample efficiency. In all cases, we first write down a factored graph as an adjacency matrix, representing the relationships between state, action, and reward. From this adjacency matrix, we then define a Factored Neural Network (Factored NN), which uses input and output masking to reflect the structure of the factored graph.

Finally, we show how to integrate this Factored NN into existing deep RL algorithms. For model-based, we use the Factored NN to learn decomposed state transitions, and then integrate this state transition model with Monte Carlo Tree Search (MCTS) (Kocsis & Szepesvári, 2006). For model-free, we use the Factored NN to learn a decomposed Q-function, and then integrate with DQN (Mnih et al., 2015). Also for model-free, we use the Factored NN to learn a decomposed policy function, and then integrate with PPO (Schulman et al., 2017). In all three cases, we demonstrate empirically that these Factored RL methods (Factored MCTS, DQN, and PPO) are able to achieve better sample efficiency than their vanilla implementations, on a range of environments.

## 2  RELATED WORK

Several methods have been proposed that exploit the structural information of a problem in the Factored MDP literature. Kearns & Koller (1999) propose a method to conduct model-based RL with a Dynamic Bayesian Network (DBN) (Dean & Kanazawa, 1989) and learn its parameters based on an extension of the Explicit Explore or Exploit ($E^3$) algorithm (Kearns & Singh, 2002). Guestrin et al. (2003) propose a linear program and a dynamic program based algorithm to learn linear value functions in Factored MDPs, and extend it to multi-agent settings (Guestrin et al., 2002). They exploit the *context specific* and *additive* structure in Factored MDP that capture the locality of influence of specific states and actions. We use the same structures in our proposed algorithms. Cui & Khardon (2016) propose a symbolic representation of Factored MDPs. Osband & Van Roy (2014) propose posterior sampling and upper confidence bounds based algorithms and prove that they are near-optimal. They show that the sample efficiency of the algorithm scales polynomially with the number of parameters that encode the factored MDP, which may be exponentially smaller than the full state-action space. Xu & Tewari (2020) extend the results to non-episodic settings and Lattimore et al. (2016) show similar results for contextual bandits. The algorithms proposed in these prior works assume a tabular (Cui et al., 2015; Geißer et al.) or linear setting (Guestrin et al., 2003), or require symbolic expressions (Cui & Khardon, 2016). We extend these ideas to deep RL algorithms by incorporating the structural information in the neural network.

Li & Czarnecki (2019) propose a factored DQN algorithm for urban driving applications. Our proposed algorithms are similar, but we extend the ideas to model-based algorithms like MCTS (Kocsis & Szepesvári, 2006), and model-free on-policy algorithms like PPO (Schulman et al., 2017). We also evaluate our algorithms on a variety of environments which encompass discrete and continuous state-action spaces. The Factored NN we propose is closely related to Graph Neural Networks (Scarselli et al., 2008; Zhou et al., 2018), which are deep learning based methods that operate on graph domain and have been applied to domains such as network analysis (Kipf & Welling, 2016), molecule design(Liu et al., 2018) and computer vision (Xu et al., 2018). Instead of explicitly embedding the neighbors of all the nodes with neural networks, we use a single neural network with masking.

NerveNet Wang et al. (2018) addresses the expressiveness of structure in an MDP, similar to our work. They focus on robotics applications and demonstrate state-action factorization with PPO. In our work, we additionally demonstrate state transition and state-reward factorization in MCTS and DQN respectively. In addition, they propose imposing a structure with Graph Neural Networks. In contrast, we propose using input and output masking without modifying the neural architecture.

Working Memory Graphs Loynd et al. (2020) uses Transformer networks for modeling both factored observations and dependencies across time steps. However, they only evaluate their method in a grid world with a single discrete action. In contrast, we demonstrate our methods on multiple environments and algorithms with factorization in state transition, state-action and state-reward relationships. In addition, our factored network is a simple extension to the existing network used to solve a problem, whereas they impose a complex network architecture.

Action masking has been used effectively to improve RL performance in multiple works (Williams & Zweig, 2016; Williams et al., 2017; Vinyals et al., 2017). We use a similar trick when applying our Factored NN to policy networks in model-free RL. However, we use both an action mask as well as a state mask to incorporate factored structure in policy networks. Our state transition networks for model-based RL also imposes masks on both input and output corresponding to current state-action and next state respectively. Wu et al. (2018) introduce an action dependent baseline in actor-critic algorithms, where a separate advantage function is learned for each action. Their method also exploits

structure available in the action space. Our method to incorporate structure is orthogonal, as we modify the policy network in actor-critic methods.

There is also a relationship between our work and the emerging intersection of reinforcement learning and causal inference, as factored graphs are are a super-set of causal graphs in the MDP setting. Lu et al. (2018) use the backdoor criterion in causal inference and variational autoencoders. Zhang & Bareinboim (2019) propose a near-optimal algorithm by taking advantage of causal inference in non-Markovian dynamic treatment regimes. Both works assume there exist unobserved confounders in the environment. We instead tackle a different problem where there are no unobserved confounders and show that there are still benefits to leverage structural information.

## 3 TERMINOLOGY

We briefly describe terminology used in this paper. We use Directed Acyclic Graphs (DAG) to represent relationships between the variables. DAGs consist of nodes and edges where the nodes correspond to random variables $X = (X_1, ..., X_d)$, and a directed edge from variable $X_i$ to $X_j$ represents that $X_i$ has an effect on $X_j$ ($X_i$ is also called the parent of $X_j$). Under Markov conditions, the joint distribution of the variables can be factored as $p(X_{1:d}) = \prod_{i=1}^{d} p(X_i | PA(X_i))$.

Consider a general Markov Decision Process (MDP) defined by $(\mathcal{S}, \mathcal{A}, \mathcal{P}, R, \rho_0, \gamma)$, where $\mathcal{S}, \mathcal{A}$ denote the state and action space respectively, $\mathcal{P}$ denotes the transition probability, $R$ represents the reward function, $\rho_0$ and $\gamma$ represent the initial distribution of the state and discount factor respectively.

In the classic RL setting, one typically assumes each state $S_{t+1}^k$ depends on the entire previous states and actions, i.e., $PA(S_{t+1}^k) = \{\{S_t^k\}_{k=1}^{|\mathcal{S}|}, \{A_t^k\}_{k=1}^{|\mathcal{A}|}\}$, where $|\cdot|$ denotes the cardinality of the space, and $PA$ denotes the parents of a node in a bayesian network. However, in many scenarios, one component of the action $A_t^k$ may only cause part of the state-space $\{S_t^k\}_{k \in C_k}$ to change, where $C_k$ is the index set of the related states of the $k^{th}$ component of the action. In other words, the parents of each state may only be a subset of the actions and previous states, i.e., $PA(S_{t+1}^k) \subsetneqq \{\{S_t^k\}_{k=1}^{|\mathcal{S}|}, \{A_t^k\}_{k=1}^{|\mathcal{A}|}\}$. Simplifying the conditional dependencies helps to construct a more accurate model, enabling us to better decompose the the dynamics and reduce complexity of the learning tasks. We assume the factored structure of the environment does not change over time.

## 4 FACTORED NEURAL NETWORK

We introduce Factored Neural Networks (Factored NN), a generic method for using knowledge from a factored graph to improve neural network predictions. The Factored NN works as follows: we start with a factored graph represented as an adjacency matrix that tells us which of our inputs influence which of our outputs. Then, we predict each output one at a time while masking all the inputs that are irrelevant for the particular output according to our factored graph. We refer to the unmodified neural network as Ordinary NN.

Figure 1 gives an example. From the factored graph on the left, we observe that output $o_1$ only depends on input $i_1$, and output $o_2$ depends on both inputs. An Ordinary NN takes $(i_1, i_2)$ as input and outputs $(o_1, o_2)$ in one go. The Factored NN instead predicts $o_1$ and $o_2$ separately using knowledge of the factored graph. When predicting $o_1$, it masks out $i_2$ and only considers relevant input $i_1$. When predicting $o_2$, it does not mask any inputs. Then $o_1$ and $o_2$ are combined into one vector so that the output form of the Factored NN is of the same form as with an Ordinary NN, so backpropagation can be done as normal.

Below, we show how to use the Factored NN in both model-based and model-free RL algorithms, using the same underlying factored structure but varying which elements of $(\mathcal{S}, \mathcal{A}, R)$ to take as input/output depending on the algorithm. Using the Multi-Cartpole environment as an example, Figure 2 illustrates how a factored graph informs the Factored NN for learning decomposed state transitions, decomposed reward functions, or decomposed policy functions. The following sections discuss the applications of these to MCTS, DQN, and PPO respectively.

The factored structure of an environment has to be manually specified. While this may seem challenging for well established benchmarks, for a real life application we still need to define the

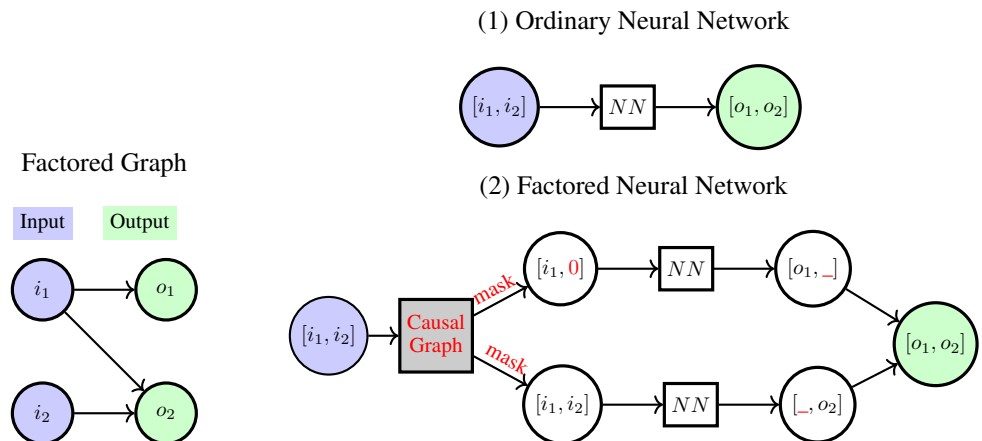

Figure 1: Illustration of Factored NN. Given the factored graph on the left, the Ordinary NN takes all the inputs and produce the corresponding outputs whereas the Factored NN masks out irrelevant input dimensions when predicting each output dimension, concatenating the outputs at the end.

MDP with state, actions and reward. Adding factorization information is relatively easy for a domain expert familiar with the details of the problem. Appendix A shows examples factored graphs for a few real world applications.

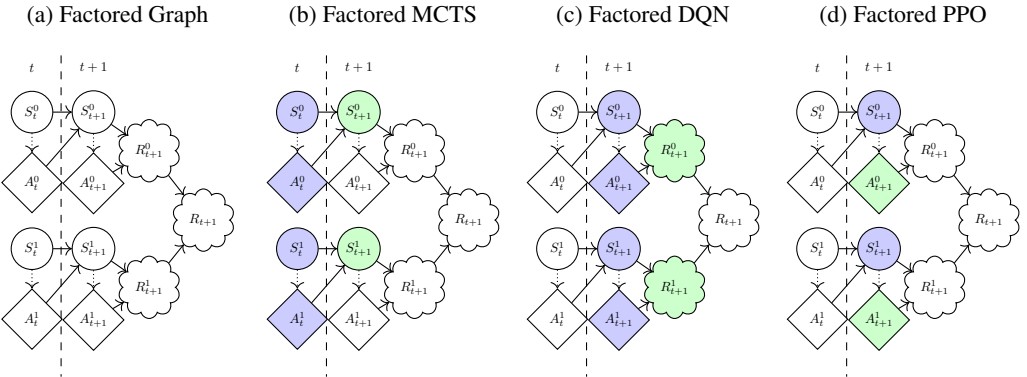

Figure 2: (a): Factored Graph for Multi-Cartpole with 2 cartpoles. $S_t^k, A_t^k, R_t^k$ for $k \in \{0, 1\}$ represent the state vector, action, and reward for each individual cartpole. (b)-(d): Graphical representation of Factored NN applications, in MCTS, DQN and PPO respectively. In each case, the input variables into the Factored NN are in blue and the outputs of the factored NN are in green.

## 5 FACTORED MCTS

In model-based RL, an environment model uses the current state and chosen actions to predict the next state, reward, and whether the episode is done or not. The more accurate the environment model, the better it can plan and the higher the rewards it can achieve - ways of improving the accuracy of environment models are therefore of interest to all model-based RL algorithms.

In order to learn an environment model more efficiently, we can construct a Factored NN that predicts next state given current state and action, according to the underlying factored structure of the problem. Taking Multi-Cartpole with factored graph displayed in Figure 2 as an example, the transition probability can be factored as $p(\mathbf{s}_{t+1}|\mathbf{s}_t, \mathbf{a}_t) = \prod_{k=1}^{d} p(\mathbf{s}_{t+1}^k|\mathbf{s}_t^k, a_t^k)$, where $d$ is the number of cartpoles, $\mathbf{s}_t^k$ and $a_t^k$ represent the state vector and the action taken for the $k^{th}$ cartpole. This efficiently reduces the complexity from a modeling perspective. The Factored NN takes input $(\mathbf{s}_t, \mathbf{a}_t)$,

decomposes accordingly and returns $\mathbf{s}_{t+1}$ as output. Figure 2b gives a graphical representation of Factored MCTS.

We can fold this Factored NN into model-based RL algorithms anywhere we use an environment model. In this work, we demonstrate using Monte Carlo Tree Search (MCTS) (Kocsis & Szepesvári, 2006) with a learned model. We implement MCTS by iterating between: 1) learning the parameters of the environment model with a gradient-based approach from existing observations; and 2) acting in the world by rolling out samples from the environment model and picking the best action using tree search.

## 6    FACTORED DQN

Model-free algorithms do not use an environment model, but rather directly learn a Q-value or policy. We can still use a Factored NN to obtain better sample efficiency, simply by specifying the relevant parts of $(\mathcal{S}, \mathcal{A}, R)$ as input/output.

In the case of DQN, we need to learn a Q-value given the current state and action and update it with:

$$Q(s_t, a_t) \longleftarrow Q(s_t, a_t) + \alpha[R(s_t, a_t) + \gamma \max_{a'} Q(s_t, a') - Q(s_t, a_t)] \tag{1}$$

When the state and action space is high-dimensional, estimating Q-value becomes computationally expensive. Adhering to the underlying factored structure, we can decompose the Q-value with Factored NN. Taking Multi-Cartpole as an illustrating example, the total reward is the summation of the individual rewards: $R(\mathbf{s}_t, \mathbf{a}_t) = \sum_{k=1}^{d} R(\mathbf{s}_t^k, a_t^k)$, where $R(s, a)$ is the reward function. We can break down the Q-value in the same way as the factored structure does not change across the episode: $Q(\mathbf{s}_t, \mathbf{a}_t) = \sum_{k=1}^{d} Q(\mathbf{s}_t^k, a_t^k)$. The Factored NN takes the state-action pair $(\mathbf{s}_t, \mathbf{a}_t)$ as the input, decomposes it into individual state-action pair $(\mathbf{s}_t^k, a_t^k)$ for each cartpole, predicts individual Q-value $Q(s_t^k, a_t^k)$ and combines them into the final $Q(\mathbf{s}_t, \mathbf{a}_t)$ which can then be updated with (1). Figure 2c gives an illustration of Factored DQN.

## 7    FACTORED PPO

Finally, we can also integrate the Factored NN into model-free algorithms that directly do policy optimization. In this work, we show how to do this with PPO (Schulman et al., 2017), an actor-critic algorithm where a policy network determines the action based on the state, and a value network predicts the episode return from the current state.

The policy network $\pi(\mathbf{a}_t|\mathbf{s}_t)$ directly optimizes the best action $\mathbf{a}_t$ given the current state $\mathbf{s}_t$. The factored structure can be used to reduce the complexity of a problem by decomposing the conditional distribution $\pi(\mathbf{a}_t|\mathbf{s}_t)$ accordingly. We can then apply the Factored NN to the policy network, by mapping only the structurally related states to the actions. In the Multi-Cartpole example, $\pi(\mathbf{a}_t|\mathbf{s}_t) = \prod_{k=1}^{d} \pi(a_t^k|\mathbf{s}_t^k)$. Factored NN takes the entire state $\mathbf{s}_t$ as input, decomposes into each individual state $\mathbf{s}_t^k$ and predicts its corresponding action $a_t^k$ for each cartpole $k$. See Figure 2d for an illustration.

## 8    EXPERIMENTS

We show experimental results for Factored MCTS, DQN, and PPO on a variety of simulation environments. In all experiments, we first define a factored graph representing the relationships among a given problem's $(\mathcal{S}, \mathcal{A}, R)$, then leverage that graph in a Factored NN to learn a policy with either MCTS, DQN, or PPO. The results below compare these FactoredRL algorithms with their vanilla counterparts. All experiments use the same hyper-parameters for Factored NN and Ordinary NN, they are reported in Appendix C. Each experiment is run on 5 different seeds.

### 8.1    FACTORED MCTS EXPERIMENTS

We experiment with Factored MCTS on two environments: Multi-Cartpole and Taxi. We chose these environments because we can easily decompose their state transitions, and they involve discrete actions which MCTS requires.

**Multi-Cartpole Experiments:** We first test using a Factored NN with MCTS on the Multi-Cartpole environment. In this environment, the agent balances multiple independent cartpoles at once, where a single cartpole is defined as per OpenAI gym (G. Brockman & Zaremba, 2016).

The state size is 4 multiplied by the number of cartpoles, as each cartpole has a state of size 4 representing its position, velocity and angle. The action taken by the agent is a binary vector representing the direction of force applied to each cartpole. The reward given to the agent is the sum of rewards for each cartpole, receiving 1/(Number of Cartpoles) if a cartpole is upright, and 0 if not.

The factored structure for this environment that we leverage in the Factored NN is that the state transitions for each cartpole are independent.

The results are displayed in Figure 3a and b. For both cases we consider, i.e. 4, and 8 cartpoles, incorporating the factored structure into the problem via the Factored NN leads to superior model prediction error and environment reward. In terms of sample efficiency, we find that Factored NN achieves the final score of the Ordinary NN in 25% and 10% of the time for 4 and 8 cartpoles respectively.

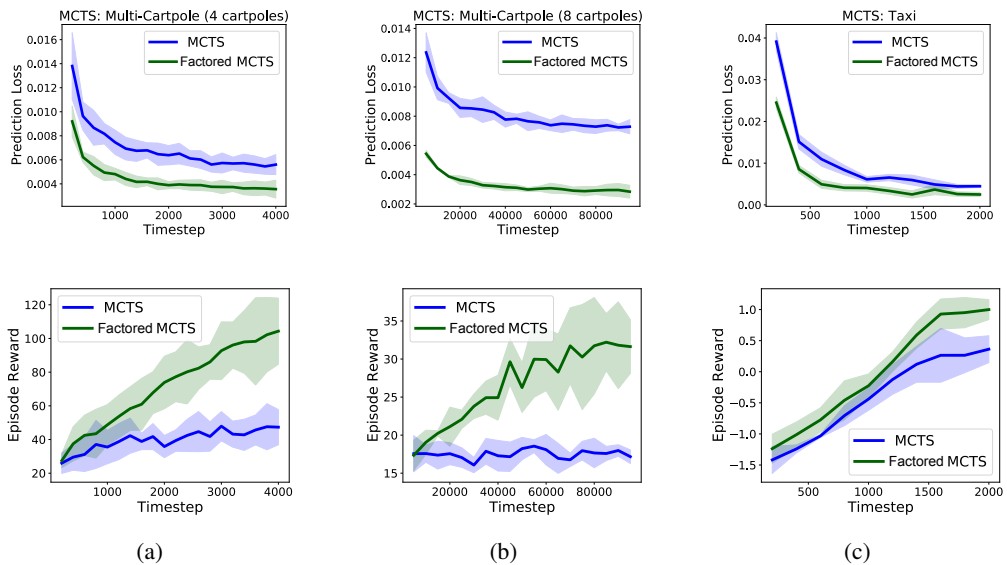

(a)                                (b)                                (c)

Figure 3: Environment model prediction loss (first row) and game score (second row) for each of the environments. Columns a and b represent environments with 4 and 8 cartpoles respectively. Column c shows the results for the Taxi environment. The solid lines show the average over 5 random seeds and the shaded area shows ± 1 standard deviation.

**Taxi Experiments:** We also test Factored MCTS on a simplified version of the Taxi environment (G. Brockman & Zaremba, 2016). In this environment there is a taxi and a passenger, and the goal is for the taxi to pick up the passenger and drop him off at a specified location. [1]

The state is given by the location of the taxi, the passenger, and the target location. The action taken by the agent is a discrete action to move up, down, left, right, or to pick up or drop off the passenger. A positive reward is given if the agent drops off the passenger in the correct location; a negative reward is given every timestep and if the taxi tries to dropoff or pickup a passenger illegally.

The factored structure for this environment that we leverage in the Factored NN is that the destination never changes and the location of the taxi and passenger only depends on a subset of actions.

The results are displayed in Figure 3c. As with the Multi-Cartpole experiments, incorporating the factored structure into the problem via the Factored NN leads to superior model prediction error and environment reward. In terms of sample efficiency, we find that Factored NN achieves the final score of the Ordinary NN in 60% of the time.

---

[1]To make the environment solvable in a reasonable amount of time using MCTS, we simplified the problem slightly by having the taxi always begin with the customer onboard and within a certain distance of its destination.

## 8.2 FACTORED DQN EXPERIMENTS

We experiment with Factored DQN on two environments: Bitflip, and Multi-Cartpole. We chose these environments because factored structure in these two environments allow us to decompose Q-values (which we cannot do in the Taxi environment), and DQN requires discrete actions.

**BitFlip Experiments:** Our BitFlip problem formulation is inspired from the example introduced by Andrychowicz et al. (2017). In this environment, the agent tries to flip bits (0 or 1) in a vector to match the values of a target vector of bits.

The state is given by two sets of $n$ bits, where one set consists of current $n$ bits and another set consists of $n$ target bits ($2n$ state). The action taken by the agent at each timestep is a discrete binary vector $\mathbf{a}$ of size $n$ where $\mathbf{a}_i = 0, 1$ means no flip and flip on the $i$-th current bit ($2^n$ possible actions). The reward given to the agent is $-1$ for each flipped bit and a positive reward of 2 for each flipped bit matched to a target bit. This environment is episodic and it ends if it reaches the maximum number of time steps ($= 3n$) or if current bits are same as target bits.

Factored NN uses the independence of the reward corresponding to each bit to estimate the respective Q-value of each bit flip. The final Q value is the sum of Q-values for each bit. Note that only sum of independent rewards are available during training time, which differentiates the problem from solving each sub-problem independently.

The results are displayed in Figure 4a. For all cases we consider, i.e., 2, 4, and 8 bits, incorporating the factored structure into the problem via the Factored NN leads to superior environment reward. In terms of sample efficiency, Factored NN achieves the asymptotic performance of the ordinary NN in 10% of the time in the 8 bit case. The results for 2 and 4 bits are presented in Appendix D

**Multi-Cartpole Experiments:** We also test Factored DQN on the Multi-Cartpole environment, with the same problem statement described in Section 8.1. Factored NN leverages the information that the reward for one cartpole is independent from other cartpoles. Again, note that only sum of independent rewards are available during training time.

The results are displayed in Figure 4b and 4c, for environments with 4 and 8 cartpoles, respectively. Factored DQN is superior in terms of mean episodic rewards. The performance gap is bigger when as we increase the number of cartpoles because the number of outputs of the Q-function grows exponentially. Factored NN reduces the complexity by estimating the Q-value of each cartpole independently. In terms of sample efficiency, Factored NN achieves asymptotic performance within 300000 timesteps while NN suffers from high-dimensional state-action pairs and does not improve its performance within 1 million timesteps.

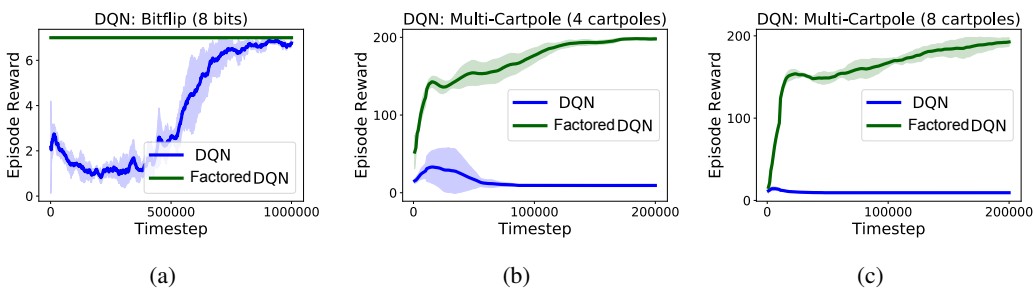

|         (a)         |         (b)         |         (c)         |

Figure 4: DQN with Factored NN vs. NN performance on Bitflip (a) and Multi-Cartpole (b, c). The solid lines show the average over 5 random seeds and the shaded area shows $\pm$ 1 standard deviation.

## 8.3 FACTORED PPO EXPERIMENTS

We experiment with Factored PPO on five environments: Bitflip, Multi-Cartpole, Half-Cheetah, Ant, and Humanoid. We chose these environments to evaluate on both discrete and continuous state-action spaces. We do not evaluate on Taxi because it has only one set of actions that are not separable by factored structure of a policy network.

**Bitflip & Multi-Cartpole Experiments:** We first test using a Factored NN with PPO on the Bitflip and Multi-Cartpole environments. These have the same setup described in Sections 8.1 and 8.2.

For BitFlip, we map the action of flipping a bit to the state of the corresponding bit, the rest of the state is masked out. For Multi-Cartpole, we map the action of controlling each cartpole with the state of the corresponding cartpole, masking out the state of other cartpoles.

We show a subset of the results in Figure 5, and the rest are given in Appendix E. For both environments, Factored PPO outperforms its vanilla implementation with respect to environment reward over the training period. In terms of sample efficiency, Factored NN achieves the final score of the Ordinary NN in 46% of the time for BitFlip with 8 bits; in 32% and 22% of the time for 4 and 8 cartpoles respectively.

**Robotics Experiments:** We also test Factored PPO on three continuous control robotics environments: Ant, Half-Cheetah, and Humanoid with a horizon of 1000 steps, as defined in PyBullet (Coumans & Bai, 2016)[2]. In these environments, the agent controls the robot joints, and its goal is to walk upright.

The state for each of these environments is slightly different, but in general state consists of joint angle and angular velocity as well as global state such as robot position and contact force with the ground. The action taken by the agent controls each joint torque. HalfCheetah, Ant and Humanoid have 6, 8 and 17 joints respectively. The reward given to the agent is positive if the robot is upright and moving. There is negative reward for using electricity and if the robot falls or tangles its legs. The Factored NN leverages the structure that the state of each robot joint maps only to the respective joint actions.

The results are displayed in Figure 6, which shows the training curves for the baseline PPO and Factored PPO. In terms of sample efficiency, Factored PPO reaches the final episode reward of baseline PPO in 35%, 61% and 64% of the time for HalfCheetah, Ant and Humanoid respectively. We observe similar results when we remove the critic in PPO, and report detailed results in Appendix E.

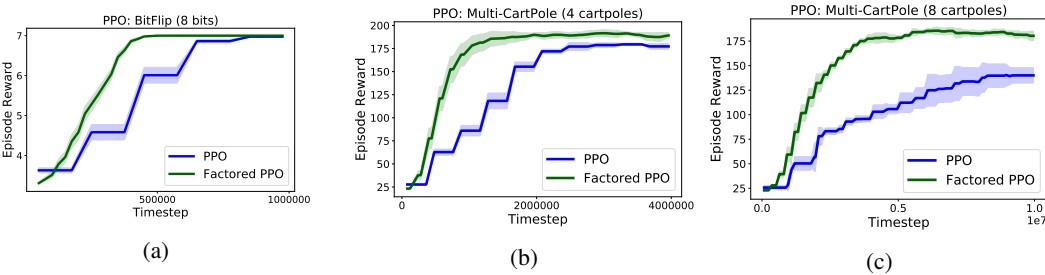

Figure 5: PPO with Factored NN vs. Ordinary NN performance on multi BitFlip with 8 bits (a), 4 (b) and 8 (c) cartpoles. The solid lines show the average over 5 random seeds and the shaded area shows $\pm 1$ standard deviation.

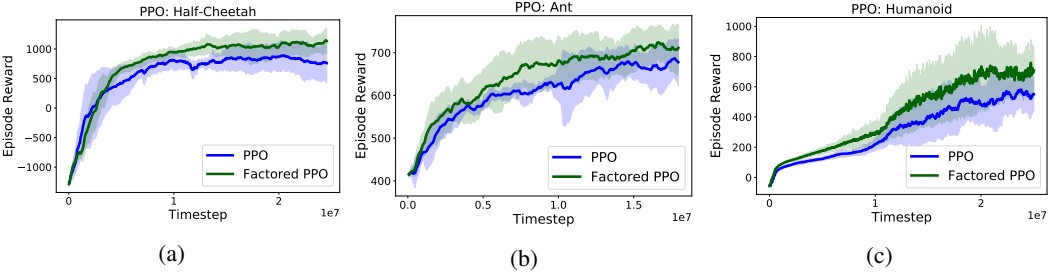

Figure 6: PPO with Factored NN vs. Ordinary NN performance on Half Cheetah (a), Ant (b), and Humanoid (c). The solid lines show the average over 5 random seeds and the shaded area shows $\pm 1$ standard deviation.

---

[2]The open source PyBullet environments are slightly different from the MuJoCo counterparts - the converged episode rewards differ. PyBullet emulates physics accurately, and is as challenging to solve as MuJoCo.

# 9 CONCLUSION AND FUTURE WORK

We demonstrate how to exploit factored structural knowledge in both model-based and model-free RL. We leverage this factored structure through a Factored Neural Network (Factored NN), which uses input and output masking to reflect the factored graph. In model-based RL, we show how to use a Factored NN to learn a state transition model, which can then be integrated into algorithms like MCTS. In model-free RL, we show how to use a Factored NN to learn a factored Q function which can be integrated into DQN, and we also show how to use a Factored NN to learn a factored policy function for use in PPO. Our method can be easily generalized to other model-based and model-free algorithms with the same idea. We have tested our FactoredRL approach on both continuous and discrete state-action spaces including bitflip, cartpole, taxi, and robotics environments, which show improved sample efficiency relative to vanilla algorithms.

The contribution of this work is in tying together factored graphs and deep reinforcement learning, showing that factored structure can improve the performance of RL algorithms. The theoretical properties of Factored NN, scalability to high dimensional inputs such as images, and alternative methods for incorporating factored structure into RL are interesting directions of future work. Critically, in this work we take as given access to an accurate factored graph, specified manually ex ante. In many environments, doing so is either time consuming, inaccurate, or outright infeasible. In future work, we will focus on factored graph discovery in online, RL settings, where the agent's actions affect the distribution of data and therefore the quality of the learned graph.

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

APPENDIX

# A    EXAMPLES OF FACTORED GRAPHS IN REAL WORLD APPLICATIONS

There are many real world RL applications where we can leverage the knowledge of a factored graph. Application domains include supply chain (Sultana et al., 2020), portfolio management (Ye et al., 2020), industrial control (Wei et al., 2017), robotics (Haarnoja et al., 2018), cellular networks (Chinchali et al., 2018), online education (Bassen et al., 2020), Internet of Things (Fraternali et al., 2020) and many more (Wu et al., 2017; Mao et al., 2019). Below we include a few examples of RL applications where we can employ our factored RL approach.

## A.1    OPERATIONS RESEARCH

Many problems in operations research, and especially supply chain, are natural environments for the use of factored graphs. This is because the historical dominant techniques, Linear and Dynamic Programming, require us to articulate problem structure in the form of a model to be solved, and this problem structure can easily be expressed in factored graphs as well. Two canonical problems with wide-ranging industrial applications can demonstrate this. First, there is the Bin Packing problem (Csirik et al., 2006; Gupta & Radovanovic, 2012; Balaji et al., 2019), which has industrial counterparts in truck packing and virtual machine packing, among others. In an RL formulation, the state is given by the amount filled in each bin, the dimensions of a new item to be packed, and the action is to place the new item in one of the bins, with a reward function for minimizing waste. The factored graph structure is key in this problem in the following way: The state of each bin in timestep t+1 is a function only of the same bin in timestep t, and the action at time t, meaning that RL can ignore spurious relationships between bin levels in predicting the state transition. Figure 7 depicts the factored graph.

**Binpacking Factored Graph**

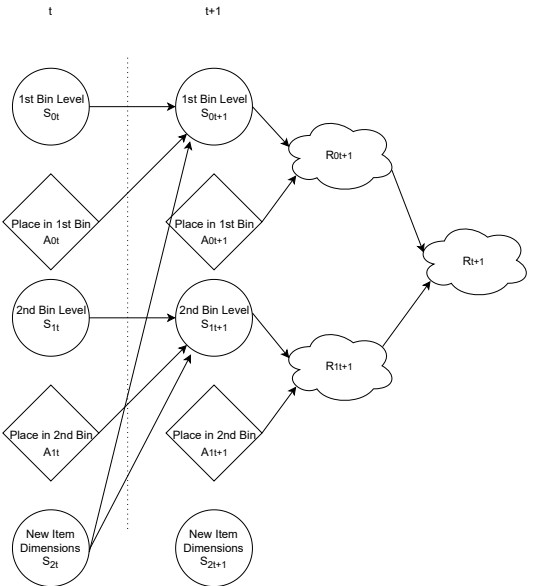

Figure 7: Bin Packing Factored Graph

Second, there is the Newsvendor problem (Balaji et al., 2019; Zipkin, 2008; Gijsbrechts et al., 2019), which has a common industrial counterpart in purchase ordering decisions (e.g. from a retailer to a vendor). In an RL formulation, the state is given by the economic parameters of the item to be ordered, the inventory on-hand, and the inventory to-arrive from the vendor in timesteps t+1, t+2, etc.,

the action is how much of the item to order, and the reward is a function of inventory on-hand and customer demand. The factored graph structure in this problem is especially helpful for simplifying the dynamics of the inventory to-arrive, since this is a linear pipeline from the vendor to the agent, i.e. the inventory that will arrive at t+1 is the same as inventory in the previous period that will arrive at t+2. Figure 8 depicts the factored graph.

**Newsvendor Factored Graph**

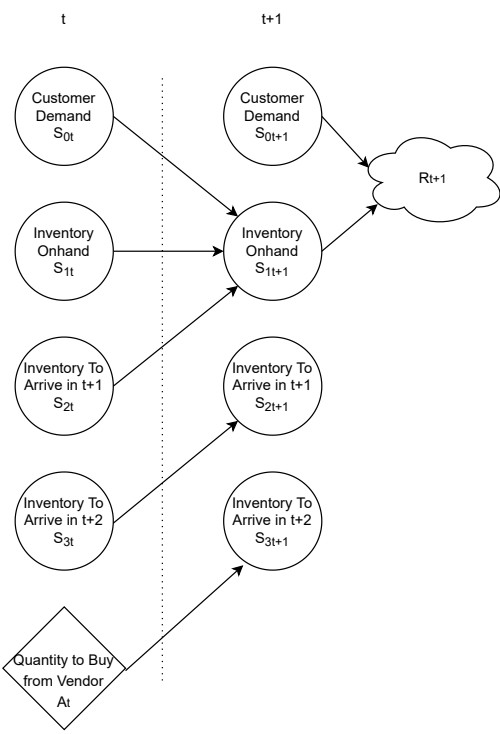

Figure 8: Newsvendor Factored Graph

## A.2 ROBOTICS

In robotics, the state may be high-dimensional, but the subspaces of the state may evolve independently of others, and only depend on a low dimensional subspace of the previous state. We have included examples of Ant, Half-Cheetah, and Humanoid in the paper with factor graph given in appendix A.4, where the transition dynamics of a robot's arms may be reasonably assumed to be independent of the transition dynamics of its legs. A similar example, we can use factored graphs for drone control with deep RL (Hodge et al., 2020).

## A.3 INDUSTRIAL CONTROL

It is common in industrial control to have several components that work together to accomplish a goal. For example in HVAC (short for heating, ventilation and air conditioning) control, the building is divided into zones each of which is controlled by a set of damper, fan and heating element (Yu et al., 2020). All the zones need to work in concert with a central air handler unit that supplies cold air. The state in this problem is the temperature of individual zones, the supply air temperature and weather conditions. The action is to set the controls of each zone. The reward is to ensure thermal comfort with minimal energy use. A state-action factored graph can be used inform the RL agent that

**Drone Control Factored Graph**

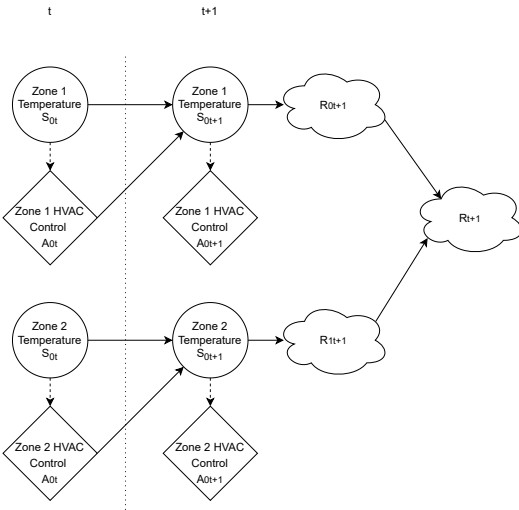

Figure 9: Drone Control Factored Graph

control of each zone is independent of each other. The reward function can also be factorized as the thermal comfort in each zone is measured independently. Figure 10 depicts the factored graph.

**HVAC Factored Graph**

Figure 10: HVAC Control Factored Graph

### A.4 PORTFOLIO MANAGEMENT

These problems (e.g. Ye et al. (2020)) generally have states of the form [stock of asset A, stock of asset B, . . . ] and action spaces of the form [buy/sell of asset A, buy/sell of asset B, . . . ]. In this scenario we have important prior knowledge about the factor graph, for example we know that the sub-action of buying/selling asset A will not influence the sub-state stock of asset B. The method would allow us to incorporate this prior knowledge into our RL Agent and improve performance. Figure 11 depicts the factored graph.

**Portfolio Management Factored Graph**

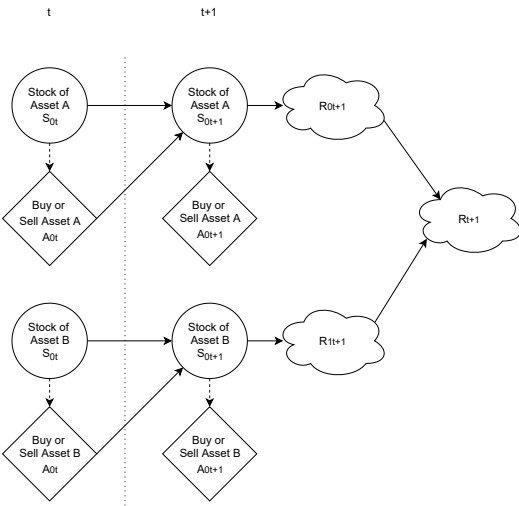

Figure 11: Portfolio Management Factored Graph

## B FACTORED GRAPHS OF ENVIRONMENTS USED IN EVALUATION

### B.1 MULTIPLE-CARTPOLE

Let $p_t^i, v_t^i$ and $\theta_t^i$ represent the position, velocity and angle for cart $i$ at time $t$ and let $A_t^i, R_t^i$ represent the force and reward for cart $i$ at time $t$, then the factored graph for this environment can be resepented by

### B.2 BITFLIP

Define $S_t^i$ to be the bit at position $i$ at time $t$, define $A_t^i$ to be the action of whether to flip the $i$-th bit at time $t$ and let $R_t^i$ represents whether the $i$-th bit equals to the $i$-th bit of the target bits. Then the factored graph for this environment can be resepented by

### B.3 TAXI

Define $p_t^{taxi}, p_t^{dest.}, p_t^{pass}$ to be the location of the taxi, target destination and passenger at time $t$ respectively, let $a_t^{move}, a_t^{pick}, a_t^{drop}$ to be the action of moving (up, down, left, right), picking up and dropping off passengers at time $t$. Then the factored graph for this environment can be resepented by

### B.4 ROBOTICS

Define $s^{global}$ to be the global features of the robot (e.g., position, contact force) and define $s_t^i$ to be the state of joint $i$ at time $t$. The action for each joint is denoted by $a_t^i$. Here we show only 3 actions

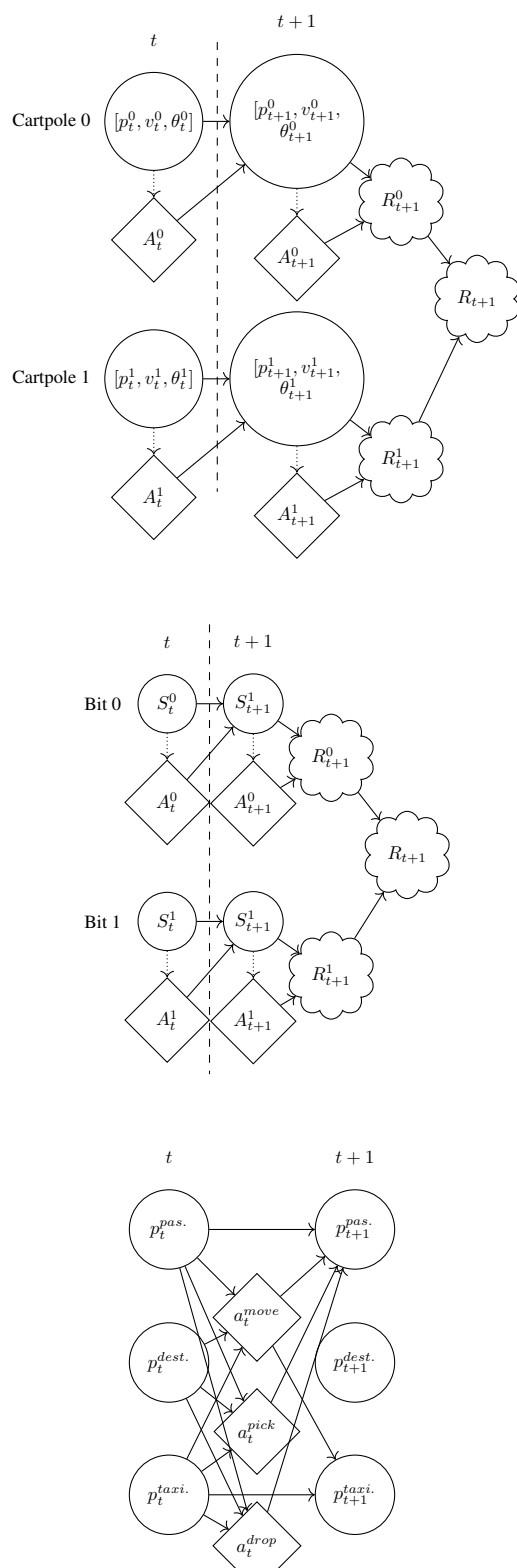

for simplicity, the graph scales similarly as we add more joints (e.g. 17 joints for the humanoid case). The factored graph for this environment can be represented by

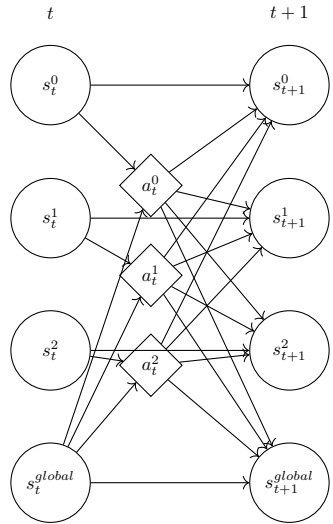

## C HYPERPARAMETERS

Below we provide the hyperparameters used for each experiment.

Table 1: Hyperparameters used for Factored MCTS Cart Pole experiments

| Hyperparameter | Value |
|---|---|
| MCTS rollouts | 50 (for 2 and 4 replicas), 200 (for 8 replicas) |
| Gamma | 0.9 |
| UCB Constant | 6.4 |
| Batch size | 64 |
| Learning rate | 0.0005 |
| Horizon | 2 |
| Warm up steps | 100 |
| Hidden layers | [100, 100] |
| Hidden layers | 1000000 |

Table 2: Hyperparameters used for Factored MCTS Taxi experiments

| Hyperparameter | Value |
|---|---|
| MCTS rollouts | 200 |
| Gamma | 0.9 |
| UCB Constant | 6.4 |
| Batch size | 16 |
| Learning rate | 0.0005 |
| Horizon | 5 |
| Warm up steps | 10000 |
| Hidden layers | [100, 100] |
| Hidden layers | 1000000 |

For PPO, we used the Ray RLlib library Liang et al. (2018). We use the default hyper-parameters in the library unless specified below. We use the hyper-parameter variable names as used in the library for ease of replicability. If some of the hyper-parameter names are unclear, please refer to the RLlib documentation for details.

Table 3: Hyperparameters used for Factored DQN experiments

| Hyperparameter | Value (Bitflip / Multi-Cartpole) |
|---|---|
| Gamma | 0.99 |
| Horizon | $3n$ ($n$: # of bits) / 200 |
| Train batch size | 128 |
| Learning rate | 0.001 |
| Initial epsilon | 0.3 |
| Target epsilon | 0.05 / 0.2 |
| Target network update frequency | 10 timesteps |
| Max time steps | 20000 |
| Hidden layers | [64, 64, 64] |

Table 4: Hyperparameters used for Half-Cheetah Factored PPO experiments. We tuned the hyperparameters to ensure convergence and improve sample efficiency.

| Hyperparameter | Value |
|---|---|
| gamma | 0.99 |
| horizon | 1000 |
| kl_coeff | 1.0 |
| num_sgd_iter | 32 |
| sgd_minibatch_size | 1024 |
| train_batch_size | 16384 |
| Learning rate | 0.0003 |
| Hidden layers | [256, 256] |
| GAE | True |
| vf_loss_coeff | 0.5 |
| grad_clip | 0.5 |
| clip_param | 0.2 |

Table 5: Hyperparameters used for Humanoid Factored PPO experiments.

| Hyperparameter | Value |
|---|---|
| gamma | 0.995 |
| horizon | 1000 |
| kl_coeff | 1.0 |
| num_sgd_iter | 20 |
| sgd_minibatch_size | 1000 |
| train_batch_size | 25000 |
| Learning rate | 0.0005 |
| Hidden layers | [256, 256] |
| lambda | 0.95 |
| clip_param | 0.2 |
| GAE | True |

Table 6: Hyperparameters used for Ant Factored PPO experiments.

| Hyperparameter | Value |
|---|---|
| gamma | 0.995 |
| horizon | 1000 |
| kl_coeff | 1.0 |
| num_sgd_iter | 20 |
| sgd_minibatch_size | 8192 |
| train_batch_size | 40000 |
| Learning rate | 0.0001 |
| Hidden layers | [256, 256] |
| clip_param | 0.2 |
| GAE | True |
| observation_filter | MeanStdFilter |

Table 7: Hyperparameters used for Humanoid Factored PPO experiments.

| Hyperparameter | Value |
|---|---|
| gamma | 0.995 |
| horizon | 1000 |
| kl_coeff | 1.0 |
| num_sgd_iter | 20 |
| sgd_minibatch_size | 1024 |
| train_batch_size | 10000 |
| Learning rate | 0.0001 |
| Hidden layers | [256, 256] |
| lambda | 0.95 |
| clip_param | 0.2 |
| GAE | True |
| clip_actions | True |
| normalize_actions | True |

Table 8: Hyperparameters used for Multi-Cartpole (8 cartpoles) Factored PPO experiments.

| Hyperparameter | Value |
|---|---|
| gamma | 0.995 |
| horizon | 1000 |
| kl_coeff | 1.0 |
| num_sgd_iter | 5 |
| sgd_minibatch_size | 1000 |
| train_batch_size | 25000 |
| Learning rate | 0.0005 |
| Max episode steps | 1000 |
| Hidden layers | [256, 256] |
| GAE | True |

Table 9: Hyperparameters used for rest of the Factored PPO experiments, including robotics experiments which exclude GAE.

| Hyperparameter | Value |
|---|---|
| gamma | 0.995 |
| horizon | 1000 |
| kl_coeff | 1.0 |
| num_sgd_iter | 20 |
| sgd_minibatch_size | 1000 |
| train_batch_size | 25000 |
| Learning rate | 0.0005 |
| Max episode steps | 1000 |
| Hidden layers | [256, 256] |

# D    ADDITIONAL DQN EXPERIMENTS

We compared Factored DQN with DQN on the BitFlip and Multi-Cartpole environments across various number of bits and cartpoles. We find that the Factored DQN consistently outperforms DQN and the performance gap is larger when the environment is harder.

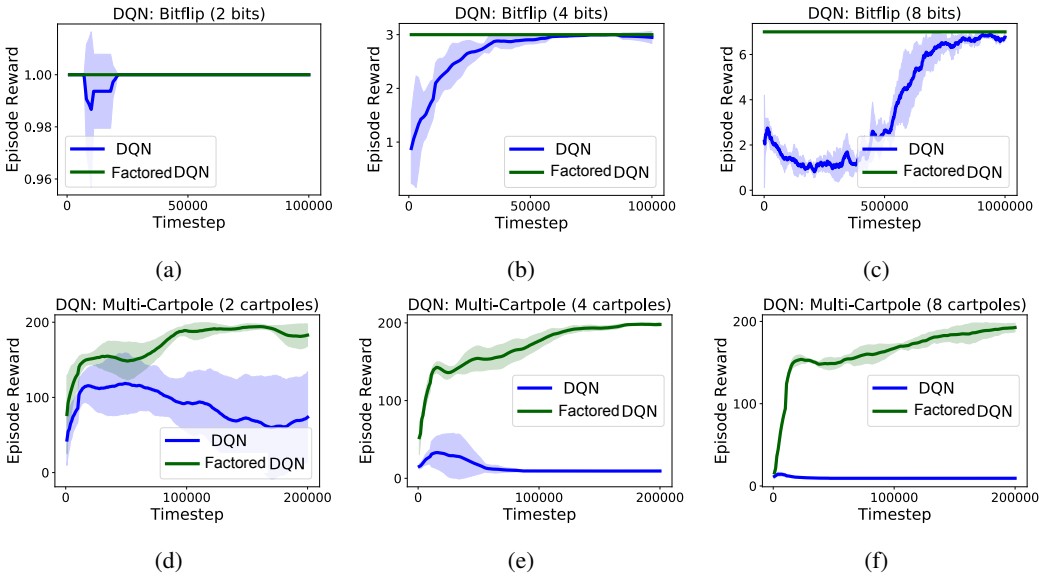

Figure 12: DQN vs Factored DQN performance on Bitflip (a,b,c) and Multi-Cartpole (d,e,f). The solid lines show the average over 5 random seeds and the shaded area shows $\pm$ 1 standard deviation.

# E    ADDITIONAL PPO EXPERIMENTS

We also compared PPO with Factored NN vs. with an ordinary NN on the BitFlip and multi-Cartpole environments with the results shown below. We find that the Factored NN algorithms perform better and that the difference increases the more complex the environment is.

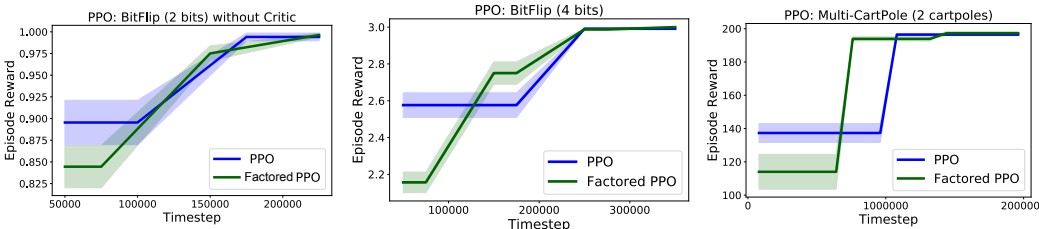

Figure 13: PPO with Factored NN vs. NN performance on Bit Flip with 2 (left) and 4 (middle) bits as well as Multi-CartPole with 2 poles (right). The solid lines show the average over 5 random seeds and the shaded area shows $\pm$ 1 standard deviation.

In addition, we report PPO results when we disable the generalized advantage estimation (GAE) Schulman et al. (2015) (Critic) and when we disable the critic altogether (without Critic).

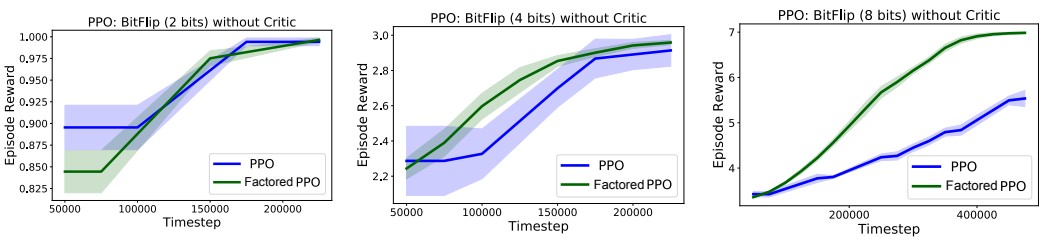

Figure 14: PPO (without a critic) with Factored NN vs. NN performance on Bit Flip with 2 (left), 4 (middle) and 8 (right) bits. The solid lines show the average over 5 random seeds and the shaded area shows $\pm$ 1 standard deviation.

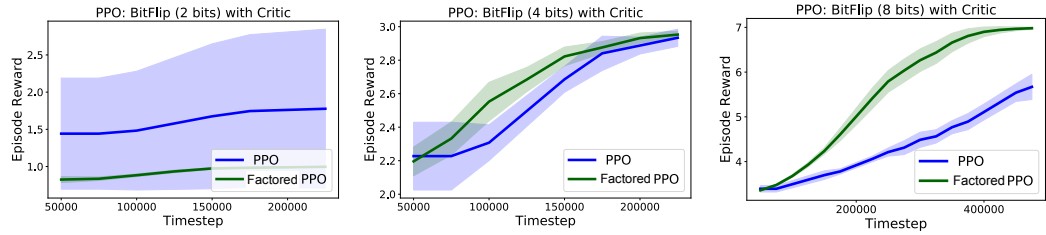

Figure 15: PPO (with a critic) with Factored NN vs. NN performance on Bit Flip with 2 (left), 4 (middle) and 8 (right) bits. The solid lines show the average over 5 random seeds and the shaded area shows $\pm$ 1 standard deviation.

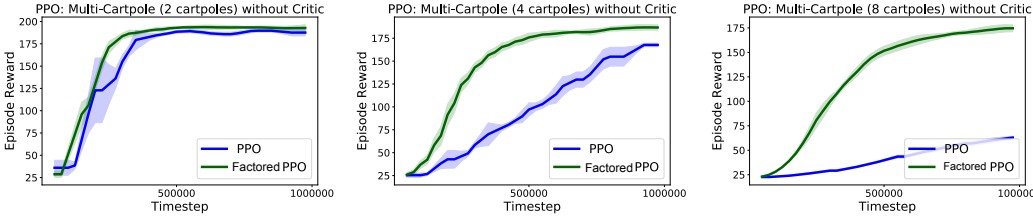

Figure 16: PPO (without a critic) with Factored NN vs. NN performance on multi Cart Pole with 2 (left), 4 (middle) and 8 (right) cartpoles. The solid lines show the average over 5 random seeds and the shaded area shows $\pm 1$ standard deviation.

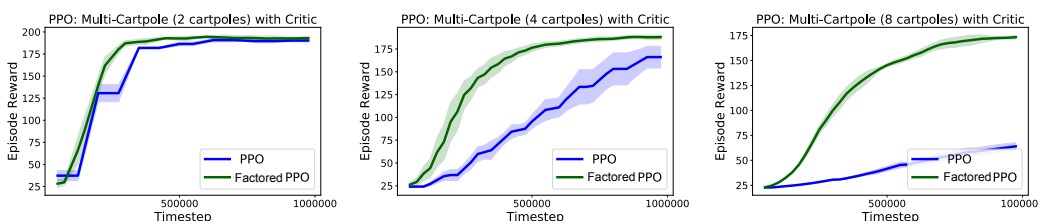

Figure 17: PPO (with a critic) with Factored NN vs. NN performance on multi Cart Pole with 2 (left), 4 (middle) and 8 (right) cartpoles. The solid lines show the average over 5 random seeds and the shaded area shows $\pm 1$ standard deviation.

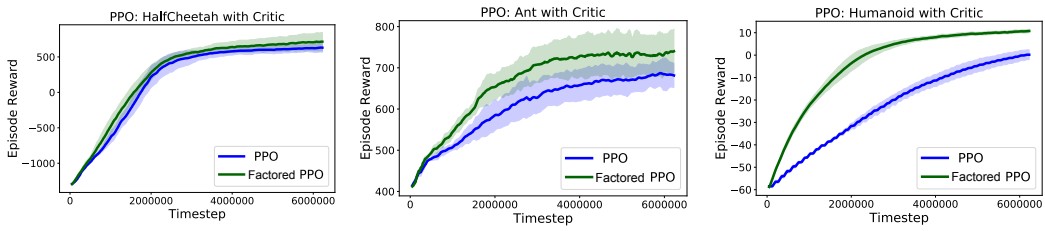

Figure 18: PPO with Factored NN vs. NN performance on Half Cheetah (left), Ant (middle), and Humanoid (right). The solid lines show the average over 5 random seeds and the shaded area shows $\pm 1$ standard deviation.

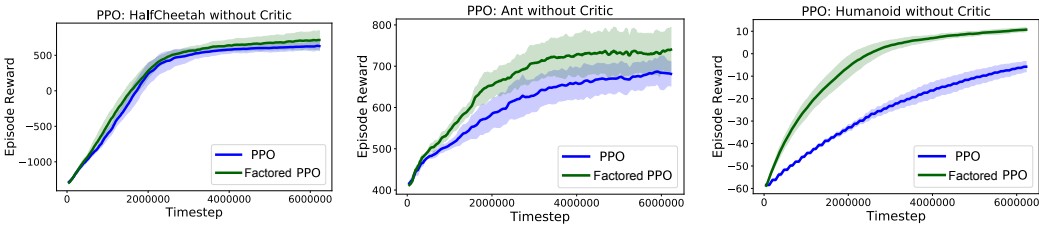

Figure 19: PPO with Factored NN vs. NN performance on Half Cheetah (left), Ant (middle), and Humanoid (right). The solid lines show the average over 5 random seeds and the shaded area shows $\pm 1$ standard deviation.

