# OpenReview forum: "FactoredRL: Leveraging Factored Graphs for Deep Reinforcement Learning"
_ICLR.cc/2021/Conference — Reject_

### Official Review · AnonReviewer3 · 2020-10-14
**Commonsense Approach to Factored MDPs**

**Rating:** 5
**Confidence:** 4

**Review:**

This paper presents a methodology for incorporating factor-graphs into model-based and model-free RL methods. The work starts by assuming access to a correct and factor graph showing the relationship between individual state factors, actions, and rewards. The authors propose to make use of this factor graph by using a Factored Neural Network - which is similar to the standard feed-forward MLP networks that would typically be used to parameterize a policy or Q-function - except that it masks out connections between input and output nodes that are not connected in the factor graph. Presumably this results in a sparser neural network which can lead to faster learning and better sample complexity. The authors demonstrate how these factored NNs can be incorporated with model-based MCTS as well as model-free DQN and PPO. In short - the algorithm remains unchanged and the only substition seems to be the Factored NN rather than a fully-connected NN. Experiments are performed on Multi-Cartpole (simultaneous control over several cartpoles), Taxi, BitFlip, and PyBullet's Ant, Half-Cheetah, and Humanoid. Each of the factored algorithms is compared with the un-factored equivalent and increased sample efficiency of learning is noted for the factored variants. The authors provide the manually-defined factor-graphs used for each of these environments in the Appendix.

On the positive side - I think the approach to creating factored-NNs to reflect the structure in the factor graph is sensible. Additionally, the results support the authors' claim that using these factored-NN can lead to greater sample efficiency. It's also nice to see that this approach - because it only modifies the underlying NN - is straightforward to integrate with many of the existing model-based and model-free algorithms.

In my opinion, the main drawback of this work is that many of the domains examined are either poor representatives of RL problems would ever be faced by an RL agent or simple enough to not require neural network function approximators. For example Multi-Cartpole is a highly-artificial environment which shows that current RL algorithms aren't equipped to simulatenously solve multiple MDPs. While this is a great motivating example - it's used extensively throughout the results. Similarly, Taxi and Bitflip are also toy domains that might not even require a neural network function approximators to solve (e.g. maybe the proper point of comparison for these domains is tabular methods built for factored MDPs?). The PyBullet environments are exceptions to this critique - but in these environments the performance improvement is not particularly notable.

More broadly, algorithms like DQN and PPO are commonly applied in environments featuring high-dimensional observation spaces such as pixel images, which don't admit easy factorization. Given that this paper is motivated by extending factorization techniques to Deep RL algorithms - the real question is "What environments admit easy factorization and require deep neural networks to solve?" It seems like the intersection of these criteria may leave vanishingly few opportunities for the community to apply the techniques in this paper.

In summary - the approach is reasonable, the results are impressive on toy domains but less impressive on realisitic domains. As pointed out in the future work, the real million dollar questions are 1) How to extract factor graphs from high dimensional pixel images and 2) How to infer latent structure of factor graphs from experience.

Minor Note - In Section 3 - PA(.) is not defined anywhere. Does it refer to the parents of a node?

Other related work - Working Memory Graphs; Loynd et al. has shown that using factored representations of state with transformer-based RL agents leads to improved sample complexity.

I have read the authors response and updated paper and appreciate the discussion of applications that admit factor graphs.

---

> ### Author Response · Authors · 2020-11-19
> **Response to AnonReviewer3**
>
> Thank you for your valuable and constructive comments.
>
> There are many real world RL applications where we do know the factored graph, and still require function approximation. Application domains include supply chain [Sultana2020], portfolio management [Yunan2020], industrial control [Wei2017], robotics [Haarnoja2017], cellular networks [Chinchali2018], online education [Bassen2020], Internet of Things [Fraternali2020], and many more [Wu2017, Hongzi2019]. We provide four factored graph examples in our overall comment. Many of these problems cannot be solved with tabular or linear factored RL algorithms due to large state-action (discrete and continuous) spaces. We agree that extending factored RL methods to vision and other high dimensional problems would be an interesting area of future research.
>
> We agree with the reviewer that the sample efficiency improvements in PyBullet environments are not as notable as in CartPole, Taxi or BitFlip environments. We note that robotic locomotion environments are challenging to improve upon, and past publications have touted marginal improvements. For example, optimistic actor critic [Ciosek2019, published in NeurIPS] shows <30% improvement in sample efficiency of Humanoid environment. In contrast, we achieve >30% sample efficiency improvements in these environments.
>
> We assumed the factor graph was known for this paper but do definitely agree with you that interesting follow-up research to this paper could be around methods that can discover the corresponding causal graphs online.  While there are known causal discovery methods [Glymour2019], causal discovery in an online MDP setting is an open problem. Prior works have developed methods for the bandit setting [Lee2018], and we hope our paper encourages causal discovery research in the full MDP setting.
>
> Thank you for pointing us to the Working Memory Graphs paper [Loynd2019]. The paper uses Transformer networks for modelling both factored observations and dependencies across time steps. Their work is indeed similar to ours. However, they only evaluate their method in a grid world with a single discrete action. In contrast, we demonstrate our methods on multiple environments and algorithms with factorization in state transition, state-action and state-reward relationships. In addition, our factored network is a simple extension to the existing network used to solve a problem, whereas they impose a complex network architecture. We will add these comparisons to the Related Work section.
>
> Yes PA(.) does refer to the parents of a node. We will clarify it in the paper.

---

### Official Review · AnonReviewer4 · 2020-10-26
**Review of "FactoredRL: Leveraging Factored Graphs for Deep Reinforcement Learning"**

**Rating:** 6
**Confidence:** 4

**Review:**

Summary:

The authors present a method by which the underlying MDP describing RL tasks can be factored into independent components in combination with Deep RL solutions to improve the efficiency and tractability of the problem.  The graph factorization of state, action and reward components is realized via an adjacency matrix that adds masking across input and output components.

They evaluate factored RL approaches for both model based and model free RL algorithms:  MCTS, DQN, PPO.  In the case of model based approach (MCTS) the state transition function is factorized.  For model-free approachs, DQN & PPO, decomposed Q-functions and policy functions are used.  These approaches are evaluated across a number of domains: Multi-Cartpole, Taxi, BitFlip and robotics tasks Ant, HalfCheetah and Humanoid from the RL Control Suite.

Overall, the paper is clear and well written.

Strengths & Weaknesses:

The authors detail the scope of the problem domain and embed their approach well in the existing literature. The scope of the problem that the authors are addressing is broad and relevant as it applies to a whole host of problems where the underlying MDP is well understood.  The novelty of the approach lies in the extension to Deep RL algorithms from prior approaches which relied on tabular or linear methods.  The authors also consider state and input/output masking in addition to action masking, an approach taken by earlier methods.  The author also seek to address and take advantage of causal inference in the fully observable RL environment setting.

The paper is divided nicely among the MCTS, DQN and PPO sections and the authors clearly describe the factored inputs and outputs in each case with the Multiple Cartpole used as an example.  The authors have also chosen a diverse set of tasks over which to evaluate and have demonstrated that there are strong gains to be realized in both sample efficiency and top-line performance when using the factored approach.

For me the greatest weakness of the paper is that there was not a great deal of insight given with respect to how scalable this approach might be in terms of other RL domains such as for instance in navigation or planning and the impact that a factored approach might have on representation learning where experience across diverse parts of the environment may be important to learn how to encode information over a diverse set of experiences.  Furthermore, the factorization algorithm needs to be  defined by hand for each task which may incur additional complexity and pose problems when the aim is to optimize for generalization in RL agents.  However the authors do state this explicitly and note that a future direction for this work could be in online discovery of factored graphs.


Other Points:

Figure 4a only contains one curve (Presumably this should be FactoredDQN rather than DQN?)


Recommendation:

I recommend a score of 6 and would be happy to see this paper accepted.   I believe the approach sets out a fruitful direction for Deep RL which could be particulary powerful if this factorization could be learned.  In these early steps I believe the authors have laid out the problem well and provided strong results to support their claim.

---

> ### Author Response · Authors · 2020-11-19
> **Response to AnonReviewer4**
>
> Thank you for your valuable and constructive comments.
>
> Yes, the factored structure of an environment has to be manually specified for each environment. While this may seem challenging for well established benchmarks, for a real life application we still need to define the MDP with state, actions and reward. Adding factorization information is relatively easy for a domain expert familiar with the details of the problem. We include example factored graphs for four problems to demonstrate the same.
>
> We assumed the factor graph was known for this paper, and interesting follow-up research to this paper could be around methods that can discover the corresponding causal graphs online.  While there are known causal discovery methods [Glymour2019], causal discovery in an online MDP setting is an open problem. Prior works have developed methods for the bandit setting [Lee2018], and we hope our paper encourages causal discovery research in the full MDP setting.
>
> We have demonstrated scalability with the Humanoid robotics environment which has 17 continuous actions, and is considered a challenging benchmark. In general, the scalability of our method relies on the function approximation provided by the neural network. We speed up the training process by reducing the stochasticity introduced by irrelevant inputs. However, we agree that factorization in domains such as navigation and planning may be challenging due to high dimensional inputs such as images, or if the structure of factored graph changes over time. These are challenging and valuable avenues of future work.
>
> The connection to representation learning and related problems such as transfer learning and multi-task learning could be interesting. We argue that factored representations may generalize better as they provide structure that is consistent with variations in the environment.  To the extent that the factored graph is consistent with the underlying causal graph, the dynamics will be consistent across variations in environment by definition.  Evidence of such representation transfer has been presented in NerveNet [Wang2018], which uses graph neural network to represent factored state and actions. However, we do not address these problems in the scope of this paper.

---

### Official Review · AnonReviewer2 · 2020-10-27
**review AnonReviewer2**

**Rating:** 6
**Confidence:** 3

**Review:**

D4RL: Datasets for Deep Data-Driven Reinforcement Learning
review:

summarization:

In this paper, the authors consider using factored neural network (NN),
instead of directly using forward NN in model-based or model-free reinforcement learning.

Pros:
1. In general the project is nice and neat.
A simple algorithm is proposed,
and the performance is improved upon existing baselines.

2. Extensions on PPO, MCTS, DQN is studied and evaluated,
making the paper quite comprehensive and the conclusion proposed more robust and convincing.

3. The paper is well written and in general easy to understand.
The figures in this paper are very helpful as well.

4. The algorithm is also applicable to high dimensional problems such as humanoid.

Cons:
1. The algorithms were not combined with existing state-of-the-art algorithms,
which includes algorithms such as SAC, TD3, MBPO.
It would be interesting to see if the proposed method can lead to state-of-the-art performance.

Misc:
1. how efficient is it to train and evaluate (is it real-time?)
2. Does it mean that the structure has to be manually specified for each environment?
3. The paper reminds me a lot of NerveNet [1],
which also considers factored or graph NN in reinforcement learning.

Summary:

In general I believe the paper is nice and neat. I tend to vote for acceptance in this case.

[1] Wang, T., Liao, R., Ba, J., & Fidler, S. (2018, February). Nervenet: Learning structured policy with graph neural networks. In International Conference on Learning Representations.

---

> ### Author Response · Authors · 2020-11-19
> **Response to AnonReviewer2**
>
> Thank you for your valuable and constructive comments.
>
> We chose to focus on validating our hypothesis over improving state-of-the-art in our experiments. We picked MCTS, DQN and PPO as three simple algorithms to test our hypothesis of factorizing the state-action space in three different ways: state transition factorization, state-reward factorization, and state-action factorization respectively. TD3 can benefit from state-reward factorization as it predicts the Q-value for a continuous action. This can be useful in domains such as portfolio or inventory management where rewards can be decomposed easily. SAC can also benefit from state-reward factorization for its Q-network, and state-action factorization for its policy network.
>
> The factored neural network requires multiple forward passes during training and evaluation. The number of forward passes is proportional to the number of input factors (or number of input masks). While it does slow down training, it can still be used for real-time applications as each forward pass takes <1s. Some of the slow down can be mitigated with batch inference and hardware acceleration.
>
> Yes, the factored structure of an environment has to be manually specified for each environment. While this may seem challenging for well established benchmarks, for a real life application we still need to define the MDP with state, actions and reward. Adding factorization information is relatively easy for a domain expert familiar with the details of the problem. We include example factored graphs for four problems to demonstrate the same.
>
> Thank you for bringing our attention to NerveNet [Wang2018]. The paper indeed addresses the expressivity of structure in an MDP, similar to our work. They focus on robotics applications and demonstrate state-action factorization with PPO. In our work, we additionally demonstrate state transition and state-reward factorization in MCTS and DQN respectively. In addition, they propose imposing a structure with Graph Neural Networks. In contrast, we propose using input and output masking without modifying the neural architecture. We will add these comparisons to our Related Work section.

---

### Official Review · AnonReviewer1 · 2020-10-30
**Nice Direction And Encouraging Results, But Not Quite Convinced About The General Appicability Of The Approach**

**Rating:** 6
**Confidence:** 3

**Review:**

This paper looks at how to build deep RL agents that can perform more efficient learning by directly leveraging the factored structure of problems when that information is given. The approach proposed by the authors is to design a neural network architecture that explicitly honors the factored structure of the problem. The authors detail a version of MCTS that can leverage independent components of the state, a version of DQN that can leverage independent components of the reward, and a version of PPO that generates actions based on independent state components. I very much like the idea of making connections with the factored MDP literature. However, as the authors point out in the conclusion, an approach that actually discovers the causal factors of the problem would be much more interesting and generally applicable. In terms of leveraging the provided structure of the problem is concerned, the experiments seem convincing that the proposed approach by the authors has value. That said, it was not clear to me how common this setting is where the agent can directly leverage known structure about independent aspects of the state and reward like this. The experiments felt a bit contrived to me, which I think at least partially reflects this inherent difficulty of  finding an environment that naturally fits the setting the authors explore. Additionally, the environments considered of very low complexity. I do not usually like to focus on this, but see it as a potential issue in this case because it is unclear to me if the architecture implications of the factored neural network considered here would scale well for example to vision problems with complex feature spaces and CNN architectures.

I am very much on the fence about this paper as I like the overall direction as a stepping stone towards a model that discovers factored state representation as well. However, I think the paper could really benefit from more evidence either theoretically or empirically that the factored neural network structure is always likely to lead to improvements when this structure is present. For example, could you say anything theoretically about the way that this leads to sample efficiency improvements during learning? Additionally, the authors should explain and potentially validate how their factored neural networks would extend to richer input streams like visual observation based environments.

After The Rebuttal: I really appreciate the rebuttal and revisions submitted by the authors. They did a good job of detailing domains of interest where their approach may be applicable. I also agree with some of the points they made about the complexity of domains that they considered in their experiments. As such, I have revised my score and now lean towards acceptance of the paper.

---

> ### Author Response · Authors · 2020-11-19
> **Response to AnonReviewer1**
>
> Thanks a lot for your thorough review and constructive comments.
>
> We assumed the factor graph was known for this paper but do definitely agree with you that interesting followup research to this paper could be around methods that can discover the corresponding causal graphs online.  While there are known causal discovery methods [Glymour2019], causal discovery in an online MDP setting is an open problem. Prior works have developed methods for the bandit setting [Lee2018], and we hope our paper encourages causal discovery research in the full MDP setting.
>
> We agree that extending factored RL methods to vision and other high dimensional problems would be a valuable and challenging area of future research. However, there are many real world RL applications where we do know the factored graph. Application domains include supply chain [Sultana2020], portfolio management [Yunan2020], industrial control [Wei2017], robotics [Haarnoja2017], cellular networks [Chinchali2018], online education [Bassen2020], Internet of Things [Fraternali2020] and many more [Wu2017, Hongzi2019]. We provide four factored graph examples in our overall comment, from our own domain knowledge. Specifying the factored graph is relatively straight forward given that we already need to define our states, actions and reward for these problems. Many of these problems cannot be solved with tabular or linear factored RL algorithms due to large state-action (discrete and continuous) spaces. These methods benefit from function approximation introduced by deep RL algorithms, even though they may not employ image based inputs.
>
> We agree some of the environments tested on were low complexity but we believe that the robotic locomotion environments are high complexity as they require a policy with up to 17 different continuous actions each step. They are also generally considered a strong benchmark for RL problems, and many seminal RL papers only test their agents on the locomotion environments (e.g. the paper that introduced soft actor critic [Haarnoja2018]). Note that we could not use MCTS or DQN for robotic environments as they are not applicable to continuous action space.
>
>
> The theoretical properties of Factored MDPs have been well studied in literature [Kearns1999, Osband2014, Xu2020]. The sample efficiency of factored RL algorithms scales polynomially to the number of “factors” in the state-action space, which can be exponentially smaller than the full state-action space. Our contribution is to extend these benefits to Deep RL algorithms (model-based, policy gradient based and Q-value based). As we only mask the relevant inputs and outputs of the RL networks, at worst the factored neural network degrades to using the unmodified network for training.

---

### Author Response · Authors · 2020-11-19
**Factored Graphs in Example Applications**

We thank the reviewers for their valuable and constructive comments. We will revise the paper based on your feedback. We have responded to each review separately.

Below we include a few examples of RL applications where we can employ our factored RL approach.

a) Operations Research — Many problems in operations research, and especially supply chain, are natural environments for the use of factored graphs.  This is because the historical dominant techniques, Linear and Dynamic Programming, require us to articulate problem structure in the form of a model to be solved, and this problem structure can easily be expressed in factored graphs as well.  Two canonical problems with wide-ranging industrial applications can demonstrate this.  First, there is the Bin Packing problem [Csirik2006, Gupta2012, Coffman2013, Agrawal2015], which has industrial counterparts in truck packing and virtual machine packing, among others.  In an RL formulation, the state is given by the amount filled in each bin, the dimensions of a new item to be packed, and the action is to place the new item in one of the bins, with a reward function for minimizing waste.  The factored graph structure is key in this problem in the following way: The state of each bin in timestep t+1 is a function only of the same bin in timestep t, and the action at time t, meaning that RL can ignore spurious relationships between bin levels in predicting the state transition.  Second, there is the Newsvendor problem [Zipkin2000, Huh2009, Rudin2014, Gijsbrechts2019], which has a common industrial counterpart in purchase ordering decisions (e.g. from a retailer to a vendor).  In an RL formulation, the state is given by the economic parameters of the item to be ordered, the inventory on-hand, and the inventory to-arrive from the vendor in timesteps t+1, t+2, etc., the action is how much of the item to order, and the reward is a function of inventory on-hand and customer demand.  The factored graph structure in this problem is especially helpful for simplifying the dynamics of the inventory to-arrive, since this is a linear pipeline from the vendor to the agent, i.e. the inventory that will arrive at t+1 is the same as inventory in the previous period that will arrive at t+2.

b) Robotics - In robotics, the state may be high-dimensional, but the subspaces of the state may evolve independently of others, and only depend on a low dimensional subspace of the previous state [Chen2020]. We have included examples of Ant, Half-Cheetah, and Humanoid in the paper with factor graph given in appendix A.4, where the transition dynamics of a robot's arms may be reasonably assumed to be independent of the transition dynamics of its legs. The state-action factored graph structure is useful for simplifying the environment model. Another example is the quadrupedal robot maze task [haarnoja2017]. In this task, a quadrupedal 3D robot needs to find a path through a maze to a target position. The reward function is a Gaussian centered at the target. The state includes the direction of the robot and the speed and the state-reward factored graph structure is helpful in this problem as the reward depends only on the speed of the robot’s motion, regardless of direction [Levine2018].

c) Industrial control - It is common in industrial control to have several components that work together to accomplish a goal. For example in HVAC (short for heating, ventilation and air conditioning) control, the building is divided into zones each of which is controlled by a set of damper, fan and heating element [Yu2020]. All the zones need to work in concert with a central air handler unit that supplies cold air. The state in this problem is the temperature of individual zones, the supply air temperature and weather conditions. The action is to set the controls of each zone. The reward is to ensure thermal comfort with minimal energy use. A state-action factored graph can be used inform the RL agent that control of each zone is independent of each other. The reward function can also be factorized as the thermal comfort in each zone is measured independently.

d) Portfolio Management - These problems [e.g. Yunan2020] generally have states of the form {stock of asset A, stock of asset B, ...other variables}  and action spaces of the form {buy/sell of asset A, buy/sell of asset B, ...other variables}.  In this scenario we have important prior knowledge about the factor graph, for example we know that the sub-action of buying/selling asset A will not influence the sub-state stock of asset B.  The method would allow us to incorporate this prior knowledge into our RL Agent and improve performance.

We will include the above examples with a problem description and the associated factored graph in the Appendix of the paper.

---

### Author Response · Authors · 2020-11-19
**References for Review Responses**

[Lee2018] Lee, S., and E. Bareinboim. "Structural causal bandits: where to intervene?." NeurIPS 2018.

[Glymour2019] Glymour, C., K. Zhang, and P. Spirtes. "Review of causal discovery methods based on graphical models." Frontiers in genetics 10 (2019): 524.

[Haarnoja2018] Haarnoja, T., A. Zhou, P. Abbeel, and S. Levine. "Soft actor-critic: Off-policy maximum entropy deep reinforcement learning with a stochastic actor." arXiv preprint arXiv:1801.01290 (2018)

[Chinchali2018] Chinchali, S., P. Hu, T. Chu, M. Sharma, M. Bansal, R. Misra, M. Pavone, and S. Katti. "Cellular Network Traffic Scheduling With Deep Reinforcement Learning." In AAAI, 2018.

[Bassen2020] Bassen, J., B. Balaji, M. Schaarschmidt, C. Thille, J. Painter, D. Zimmaro, A. Games, E. Fast, and J. C. Mitchell. "Reinforcement Learning for the Adaptive Scheduling of Educational Activities." In CHI 2020.

[Fraternali2020] F. Fraternali, B. Balaji, D. Sengupta, D. Hong, and R. K. Gupta. 2020. Ember: energy management of batteryless event detection sensors with deep reinforcement learning. In SenSys 2020.

[Yunan2020] Y. Yunan, H. Pei, B. Wang, P. Chen, Y. Zhu, J. Xiao, and B. Li. "Reinforcement-learning based portfolio management with augmented asset movement prediction states." In AAAI 2020.

[Wei2017] Wei, T., Y. Wang, and Q. Zhu. "Deep reinforcement learning for building HVAC control." In Proceedings of the 54th Annual Design Automation Conference 2017.

[haarnoja2017] Haarnoja, T., et al. "Reinforcement learning with deep energy-based policies." arXiv preprint arXiv:1702.08165 (2017).

[Sultana2020] Sultana, N. N., H. Meisheri, V. Baniwal, S. Nath, B. Ravindran, and H. Khadilkar. "Reinforcement Learning for Multi-Product Multi-Node Inventory Management in Supply Chains." arXiv preprint arXiv:2006.04037 (2020).

[Kearns1999] Kearns, M., and D. Koller. "Efficient reinforcement learning in factored MDPs." In IJCAI. 1999 (tel:7407471999).

[Osband2014] Osband, Ian, and Benjamin Van Roy. "Near-optimal reinforcement learning in factored mdps." In Advances in Neural Information Processing Systems 2014.

[Xu2020] Xu, Z,, and A. Tewari. "Near-optimal Reinforcement Learning in Factored MDPs: Oracle-Efficient Algorithms for the Non-episodic Setting." arXiv preprint arXiv:2002.02302 (2020).

[Wang2018] Wang, T., R. Liao, J. Ba, and S. Fidler. "Nervenet: Learning structured policy with graph neural networks." In ICLR 2018.

[Loynd2019] Loynd, R., R. Fernandez, A. Celikyilmaz, A. Swaminathan, and M. Hausknecht. "Working Memory Graphs." arXiv preprint arXiv:1911.07141 (2019).

[Hongzi2019] Mao, H., P. Negi, A. Narayan, H. Wang, J. Yang, H. Wang, R. Marcus et al. "Park: An open platform for learning-augmented computer systems." NeurIPS (2019)

[Wu2017] Wu, C., A. Kreidieh, K. Parvate, E. Vinitsky, and A. M. Bayen. "Flow: Architecture and benchmarking for reinforcement learning in traffic control." arXiv preprint arXiv:1710.05465 (2017).

[Ciosek2019] Ciosek, K., Q. Vuong, R. Loftin, and K. Hofmann. "Better exploration with optimistic actor critic." In NeurIPS 2019.

[Coffman2013] Coffman, E. G., J. Csirik, G. Galambos, S. Martello, and D. Vigo. "Bin packing approximation algorithms: survey and classification." In Handbook of combinatorial optimization, Springer New York, 2013.

[Csirik2006] Csirik, J., D. S. Johnson, C. Kenyon, J. B. Orlin, P. W. Shor, and R. R. Weber. "On the sum-of-squares algorithm for bin packing." Journal of the ACM (JACM) (2006).

[Gupta2012] Gupta, V., and A. Radovanovic. "Online stochastic bin packing." arXiv preprint arXiv:1211.2687 (2012).

[Agrawal2014] Agrawal, S., and N. R. Devanur. "Fast algorithms for online stochastic convex programming." In Proceedings of the twenty-sixth annual ACM-SIAM symposium on Discrete algorithms, Society for Industrial and Applied Mathematics, 2014.

[Zipkin2008] Zipkin, P.. "Old and new methods for lost-sales inventory systems." Operations Research 2008.

[Huh2009] Huh, W. T., G. Janakiraman, J. A. Muckstadt, and P. Rusmevichientong. "Asymptotic optimality of order-up-to policies in lost sales inventory systems." Management Science 2009.

[Rudin2014] Rudin, C., and G. Vahn. "The big data newsvendor: Practical insights from machine learning." (2014).

[Gijsbrechts2019] Gijsbrechts, J., R. N. Boute, J. A. Van Mieghem, and D. Zhang. "Can deep reinforcement learning improve inventory management? performance on dual sourcing, lost sales and multi-echelon problems." Performance on Dual Sourcing, Lost Sales and Multi-Echelon Problems (July 29, 2019) (2019).

[Yu2020] Yu, L., Y. Sun, Z. Xu, C. Shen, D. Yue, T. Jiang, and X. Guan. "Multi-agent deep reinforcement learning for HVAC control in commercial buildings." IEEE Transactions on Smart Grid (2020).

---

### Decision · Program_Chairs · 2021-01-07
**Final Decision**

**Decision:**

Reject

**Comment:**

The paper introduces variants of RL algorithms that can consume factored state representations. Under the assumption that actions only affect a few factors, these factored RL algorithms can learn more efficiently than their vanilla counterparts. Learning a factored dynamics model (to be used in a model-based algorithm) or representing factorized action-selection policies (to be optimized by a model-free RL algorithm) make intuitive sense in the problem settings that the paper considers. However, the paper should clarify the implicit assumptions being made about how the reward decomposes across factors. For instance, the factored DQN approach seems to require a linear reward decomposition across the factors.
The factored DQN approach is also reminiscent of the Hydra algorithm on MsPacMan (https://papers.nips.cc/paper/2017/file/1264a061d82a2edae1574b07249800d6-Paper.pdf Section 4.2) which assigns an RL agent to each factor ("ghost" in MsPacMan) to learn a factor-specific Q-function. The linear aggregator that they use is identical to the factored DQN in this paper.

The reviewers all rate the paper as borderline. All reviewers suggest that being able to learn the factor graph (or at least parts of it) will greatly widen the scope of applications where the approach can be fruitfully applied -- the paper acknowledges this as a compelling line of future work. The biggest weakness is originality -- the core message of the paper is just that, where factored representations of state/actions exist RL algorithms must use it. This is not a surprising or novel message. The paper advocates for incorporating the factorization information in the most straightforward way (state-masking, followed by action concatenation). Simple-in-retrospect is usually an excellent feature of an algorithm, not a bug; however, the proposal is literally the first idea a reader will likely think of. It might help to explore other ways of incorporating factorization information (e.g., rather than parameter sharing, have a separate network for each factor; rather than masking, have different width input layers to consume different number of parents in the DAG; etc.) and verifying that they are inferior to factored NN.